# Analysis of Non-Minimum Phase System for AC/DC Battery Charger Power Factor Correction Converter

**Mahmoud Nassary** * , **Enric Vidal-Idiarte** and **Javier Calvente**

Departament d'Enginyeria Electrònica, Elèctrica i Automàtica, Universitat Rovira i Virgili,
43007 Tarragona, Spain; enric.vidal@urv.cat (E.V.-I.); javier.calvente@urv.cat (J.C.)
* Correspondence: mahmoud.nassary@urv.cat

**Abstract:** Electric mobility is nowadays one of the more important trends regarding pollution reduction and global warming due to fuel consumption. Big efforts are done in order to develop efficient and reliable power electronic systems for electric vehicles. In two stage on board-battery chargers, one way of improving efficiency is by means of ensuring the DC-DC isolated converter always operates in the nominal input/output voltage ratio, that could be achieved with a variable DC-link operation. In this paper, a four-switch buck-boost based AC/DC converter is deeply analyzed in order to improve its dynamic performance, the power factor and the total harmonic distortion. The converter suffers from a non-minimum phase characteristic in different input–output transfer functions, which reduces the closed-loop bandwidth of the system. Therefore, after a deep converter analysis has been done, different solutions have been evaluated and tested. Finally, a control to different output transfer functions of the converter become minimum phase, which allows us to increase the system bandwidth and, consequently, high power factor, low harmonics distortion, single control structure and fast dynamics for wide output voltage range are achieved.

**Keywords:** battery charger; PFC; electric vehicle; stability analysis; small signal modeling

## 1. Introduction

Recently, transportation electrification had rapid growth beside the grid integration of efficient electric vehicle (EV), which is becoming exponentially essential. Geographically, China is on the top of the EV market, then Europe on the second place, followed by the United States. Automakers continue to speed up their EV production to comply with global and specific country regulations to reduce greenhouse gas effects. Until 2022, it is expected that more than 500 models of EVs will exist worldwide, making electric cars more accessible and attractive to larger amounts of people.

Due to battery technology price decreases, longer driving range and availability of charging infrastructure EV sales grew from 450,000 in 2015 to 2.1 million in 2019. In addition, further increases in the market sales to 8.5 million by 2025, 26 million by 2030, and 54 million by 2040 are expected. This growing market, from the power electronics point of view, puts a huge responsibility on research and development (R&D) labs to find highly efficient, low cost, and low energy density converters for the on-board battery charging (OBC) and inverter, as well for the motor drive.

In SAE Standard J1772, the system of EV battery charger is categorized into main three categories [1]. Table 1 reviews the three infrastructure and charging power levels. Level-1 is usually used for home charging overnight when the vehicle is parking. This slow charging level takes from 8–11 h to make the battery fully charged, and this is considered an OBC type. The level input voltage can be supplied with outlet 120/240 VAC without the need for dedicated facilities. The cost of infrastructure is between $500–$880 [2]. Level-2 is considered semi-fast charging [3]. A dedicated outlet is required where it can handle a higher current, and it requires electric vehicle supply equipment (EVSE). Level-2 is

considered an on-board type. A higher current can be handled in this level, which makes it more attractive than Level-1, wherein the charging time is reduced.

However, its cost is up to $3000 due to the needs of a special outlet and EVSE [4]. Level-3 is called the fast battery charger; nevertheless, it requires a special bulky infrastructure, consequently it can't be on-board, and it should be an off-board type. Therefore, its infrastructure cost is significantly high in comparison with the other two levels, between $30,000 and $160,000. Its input must be a three-phase system, so it can handle a huge amount of power to charge the battery in 20–50 min.

**Table 1.** EV battery charger levels [1].

| Power Level Type | Charger Location | Used in | Outlet | Expected Power (kW) | Charging Time | Vehicle Capacity (kWh) |
|---|---|---|---|---|---|---|
| Level-1 120/230 Vac | On-board 1-phase | Home, office parking | Typical | 1.9 | 11–36 h | 3–50 |
| Level-2 120/230 Vac | On-board 1/3-phase | Public or private outlets | Special | 19.2 | 2–3 h | 3–50 |
| Level-3 240 Vac/600 Vdc | Off-board 3-phase | Commercial, like filling stations | Special | 50–100 | 20–50 m | 20–50 |

A two-stage topology with AC–DC power factor correction (PFC) converter as the first stage then followed by an isolated DC–DC converter is nowadays the most used topology of OBCs in EV [5]. They are well known as power-factor correction converters or input current reshapes [6,7]. An extensive review of PFC topologies is addressed in [8]. Sensorless predictive control for the versatile AC–DC buck–boost converter operating as PFC is proposed in [9]. Many researchers have tried to minimize the electrolytic capacitors or eliminate them totally to increase the converter energy density aiming for low-cost solutions PFC in [10–12]. It has a simple controller with the capability of universal input voltages from 85–240 V. This approach has many advantages in the OBC application; however, the output current suffers from high ripple, which could affect badly in the next DC-DC stage or the battery itself. A detailed discussion had been made in [13] to make a comparison between different single stage isolated AC-DC OBC for EV. This paper focused on Level-2 OBC where the input voltage maximum is 240 VAC. Main important features of the OBC are higher energy density, lower cost, and less magnetic losses as well.

Nowadays, the battery pack technology has input voltage from 200 V when the battery is fully discharged and 400 V when the battery is fully charged. In Level-2, OBC used the outlet voltage between 120/230 VAC RMS. Due to that, boost converter topologies are usually used to regulate DC-link voltage at 400 V. The DC-DC converter is only responsible to charge the battery. However, DC-DC isolated stages, resonant LLC or phase shifted full bridge topologies, decrease their efficiency due to the battery voltage variation. A buck–boost converter as PFC can adapt DC-link voltage to maintain the input/output voltage ratio constant, guaranteeing the optimal efficiency point. Nevertheless, single-switch topologies with boost and buck functions, such as SEPIC, buck-boost and Cuk converters, can comply with the battery pack voltage requirement, but the main disadvantage of those single-switch converters is the components voltage and current stresses. Thus, they are not preferable to be used in high power application where high efficiency and reliability are needed.

Figure 1 shows four switches AC/DC boost/buck converter. This converter has a capability of being used as PFC with boost and buck feature in the same double line frequency cycle, wherein the first two switches ($Q_1$-D1) are working as a boost and the other two switches ($Q_2$-D2) are working as a buck. This topology has many other applications like telecommunication system, fuel-cell systems [14], power supply equipment's [15], and radio frequency amplifier applications [16]. The main disadvantage of this topology, in open loop configuration at boost mode, the duty cycle to input current and duty cycle to output

voltage have a right half plan zero (RHPZ), which could become unstable in closed-loop. An intuitive solution for this problem is decreasing the system dynamics by minimizing the current control loop bandwidth, and this will deteriorate the power factor, which is a vital factor in the OBC. To overcome this problem, many publications addressed this by using different control techniques such as current programmed control [17], discontinuous capacitor voltage mode control [18], voltage mode control with two different proportional–integral–derivative (PID) controllers [19], and input voltage feed-forward controllers [20]. The prior arts didn't mention the wide range of the output voltage that is necessary for the battery back specification.

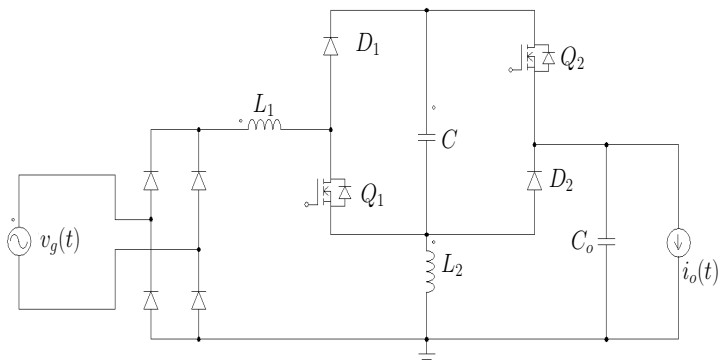

**Figure 1.** AC/DC single stage boost/buck power factor correction (PFC) converter.

In [21], AC-DC CCM operation OBC was discussed to have a wide input and output voltage range with PFC controller capability. The main disadvantage of this topology as discussed earlier that it has RHPZ in the duty cycle to input inductor current transfer function $Gi_{L1}d(s)$, which makes the system non-minimum phase. The paper claimed that it could achieve the minimum phase system by only using a simple proportional–integral (PI) control. A highly distorted input current appears on the experimental graphs in the paper. On top of that, the displacement factor is obvious as well. In addition, the paper ignored the high resonance peak that appears in the current control loop which could worsen the internal dynamics of the whole system.

The paper contribution is solving the RHPZ problem of the previous converter. A proper modeling should be done to overcome all the previous disadvantages and increase the OBC reliability and PF. The paper is focusing on a detailed converter modeling for both boost and buck modes. This will help in solving the non-minimum problem in the converter. In addition, the paper proposes reducing or eliminating completely the resonance peak in the current control loop to maintain a high reliable system, which implying a trade off between the efficiency and system reliability. The paper is organized as sections and subsections. The first section discusses the converter modeling and supporting that with numerical analysis to show the main disadvantage of this converter. Additionally, the analysis of the obtained small-signal model of the converter allows us to obtain a minimum phase condition for the input/output transfer functions. The second section discusses the effect of introducing a small low power snubber circuit to make the transfer functions of duty cycle to input current and duty cycle to output voltage minimum phase. The third section discusses the design of a simple controller to maintain a high power factor (PF) and distortion free input current. The feasibility of the proposal is validated by means of simulation in the last section.

## 2. Circuit Operation

Figure 2 shows the converter operation modes that will be discussed in detail in this section. The analyzed converter has been design to operate in continuous conduction mode (CCM), which means the ripple current is significantly smaller than the average current thus, the peak current decreases in comparison with the operation in discontinuous conduction

mode (DCM). In CCM, it has only two operating states; when a controlled switch is off the corresponding diode is forward biased. In the converter analysis, capacitors, inductors, diodes and MOSFETs are assumed ideal, neither equivalent series resistance (ESR) nor other parasitic effects are considered. The input voltage waveform is a rectified sine wave which obtained by a passive full bridge rectifier of the grid input sine voltage.

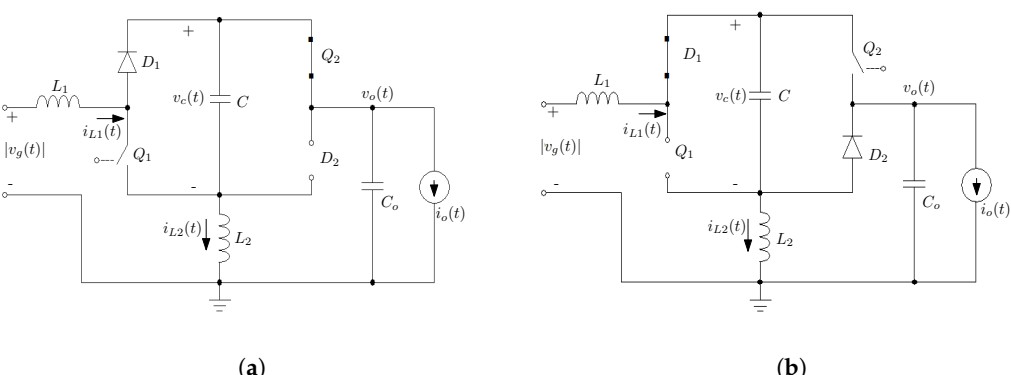

(**a**)                                                (**b**)

**Figure 2.** Simplified model for different mode: (**a**) Boost mode. (**b**) Buck mode.

Transferring the power from the input port to the output port is done by two different modes, and it depends on the needed input and output voltages as follows:

**Boost mode**: Figure 2a shows a simplified model for the converter operation in boost mode. In this operation mode, the input voltage is lower than the output voltage and the two main switching elements are the switch $Q_1$ and diode D1. Meanwhile, the switch $Q_2$ is being on and the diode D2 is being off for the whole period. Inductor $L_1$ is the main energy transfer element for the boost mode. As mentioned, the converter operates in CCM with two different states. The first state (State -I), when switch $Q_1$ is on and diode D1 is off (reverse biased). At the same period, the intermediate capacitor $C$ starts discharging. In the second CCM state (State -II), when the switch $Q_1$ is off and diode D1 is on (forward biased), Inductor $L_1$ starts to discharge its energy to the output load and charge the two capacitors: intermediate $C$ and output capacitor $C_o$.

**Buck mode**: Figure 2b shows the simplified model for the converter operation in buck mode. In this operation mode, the input voltage is higher than the output voltage. The two main converter switching elements are switch $Q_2$ and diode D2. Meanwhile, the switch $Q_1$ is being off and the diode D1 is being on for the whole period. Inductor $L_1$ is the main energy transfer element for the buck mode as well. In State-I, when the switch $Q_2$ is on and diode D2 is off, inductor $L_1$ starts charging by the input inductor current $i_{L1}$. At the same time, inductor $L_2$ and the intermediate capacitor $C$ are discharging to the output load through switch $Q_2$. The load current $i_o(t)$ is the sum of the input inductor current $i_{L1}$ and the second inductor current $i_{L2}$. In State-II, when the switch $Q_2$ is off and diode D2 is on, inductor $L_1$ starts to discharge its energy to the output load. At same time, inductor $L_2$ and intermediate capacitors $C$ are charging.

## 3. Converter Analysis

In order to analyze the converter dynamics, different state-space equations for the different modes of operation are obtained. Later, respective small-signal models are also derived. The converter has two different modes: boost and buck modes. The following analysis will be carried out for each mode as follows:

### 3.1. State-Space Averaging and Small-Signal Model Analysis in Boost Mode

In CCM, the switching cycle has two states. The duty cycle $d_1$ is the fraction of time in which the controlled switch is on. The state-space averaging model is a weighted sum of the state equations in each switching state. The weights are the corresponding fractions of time. The state variables of the converter are the input inductor current $i_{L1}(t)$, second inductor current $i_{L2}(t)$, intermediate capacitor voltage $v_c(t)$ and finally the output capacitor voltage

$v_o(t)$ as shown in Figure 2a. Assume $v_g(t)$ is positive as a rectified sine waveform. After analyzing different converter switching states, average state-space equations are obtained to represent the converter operation in boost mode as follows:

$$
\begin{aligned}
\frac{di_{L1}(t)}{dt} &= \frac{v_g(t) - v_o(t) + d_1(t)v_c(t)}{L_1}, \\
\frac{di_{L2}(t)}{dt} &= \frac{v_o(t) - v_c(t)}{L_2}, \\
\frac{dv_c(t)}{dt} &= \frac{i_{L2}(t) - d_1(t)i_{L1}(t)}{C}, \\
\frac{dv_o(t)}{dt} &= \frac{i_{L1}(t) - i_{L2}(t)}{C_o} - \frac{v_o(t)}{R_o C_o},
\end{aligned}
\tag{1}
$$

where $v_g(t)$ is the input voltage, $v_o(t)$ is the output voltage, $d_1(t)$ is the duty cycle for the boost mode, $v_c(t)$ is the voltage across the intermediate capacitor $C$, $i_{L1}(t)$ is the current through the inductor $L_1$, $i_{L2}$ is the current in the inductor $L_2$ and $R_o$ is the output resistance calculated from the output power $P_o$ and average output voltage $V_o$.

The above equations in (1) are nonlinear due to the multiplication of time-varying quantities. Most of the ac circuit analysis such as Laplace transform and Bode plot are not suitable for nonlinear systems. So, linearization is required for Equation (1). On top of that, linearization is a linear approximation of a nonlinear system that is valid in a small region around an operating point. All the variables in (1) are averaged without switching ripple. The steady-state values of the converter variables are obtained from (1) after equating to zero the different differential equations, and the variations in $v_g(t)$ are assumed to be much slower than the converter dynamics, so the converter always works close to an equilibrium point where it can be represented as follows:

$$
\begin{aligned}
V_C &= V_o, \\
D_1 &= 1 - \frac{V_g}{V_o}, \\
I_{L1} &= \frac{V_o^2}{R_o V_g}, \\
I_{L2} &= -\frac{-V_o^2 + V_g V_o}{R_o V_g}.
\end{aligned}
\tag{2}
$$

Then, the converter variables can be decomposed as:

$$
\begin{aligned}
v_g(t) &= V_g, \\
d_1(t) &= D_1 + \Delta d_1(t), \\
i_{L1}(t) &= I_{L1} + \Delta i_{L1}(t), \\
i_{L2}(t) &= I_{L2} + \Delta i_{L2}(t), \\
v_c(t) &= V_C + \Delta v_c(t), \\
v_o(t) &= V_o + \Delta v_o(t),
\end{aligned}
\tag{3}
$$

where variables $V_g, D_1, I_{L1}, I_{L2}, V_C, V_o$ express the dc equilibrium point and the other terms $\Delta d_1(t), \Delta i_{L1}(t), \Delta i_{L2}(t), \Delta v_c(t), \Delta v_o(t)$ represents small variation around it.

In this converter, the voltage output $V_o$ should be controlled for regulating its average, and the input current $i_{L1}(t)$ is controlled to reshape the line current. Thus, two transfer functions must be calculated: the duty cycle to input inductor current $Gi_{L1}d\_u(s)$ and the duty cycle to output voltage $Gvd\_u(s)$. Linearizing after substituting (3) in (1) and transform differential equations to the Laplace domain allow to obtain:

$$
\begin{aligned}
Gi_{L1}d\_u(s) &= \frac{A_1\left(s^3 + A_2\,s^2 + A_3\,s + A_4\right)}{Den\_u(s)}, \\
Gvd\_u(s) &= \frac{A_v{}^2\,V_g\left(R_o\left(A_v\,C\,L_2\,s^2 + 1\right) - A_v{}^2\,s\left(L_1 - L_2\left(\frac{1}{A_v} - 1\right)\right)\right)}{Den\_u(s)},
\end{aligned}
\tag{4}
$$

where,

$$
\begin{aligned}
Den\_u(s) &= B_4 s^4 + B_3 s^3 + B_2 s^2 + B_1 s + R_o, \\
A_1 &= A_v{}^2\,C\,Co\,L_2\,Ro\,Vo, \\
A_2 &= \frac{C - Co\,(A_v - 1)}{C\,Co\,Ro}, \\
A_3 &= \frac{C_{op} - \frac{L_2}{Ro^2}(A_v - 1)}{C\,Co\,L_2}, \\
A_4 &= \frac{2}{C\,Co\,L_2\,Ro}, \\
B_1 &= A_v\left(L_s + L_2\left(\frac{1}{A_v} - 2\right)\right), \\
B_2 &= R_o\left(C_{op}\,A_v{}^2\,L_s - C_o\,L_2\,(2\,A_v - 1)\right), \\
B_3 &= A_v{}^2\,C\,L_1\,L_2 \\
B_4 &= A_v{}^2\,C\,L_1\,L_2\,C_o\,R_o, \\
L_s &= L_1 + L_2, \\
A_v &= \frac{V_o}{V_g}, \\
C_{op} &= C + C_o.
\end{aligned}
$$

### 3.2. State-Space Averaging and Small-Signal Model Analysis in Buck Mode

Figure 2b shows a simplified model of the converter in buck mode. The same procedure of the boost mode modeling will be followed to model the transfer function for the buck mode. The average state-space model in the buck mode can be expressed as follows:

$$
\begin{aligned}
\frac{di_{L1}(t)}{dt} &= \frac{v_g(t) - v_o(t) - v_c(t)(1 - d_2(t))}{L_1}, \\
\frac{di_{L2}(t)}{dt} &= \frac{v_o(t) - v_c(t)d_2(t)}{L_2}, \\
\frac{dv_c(t)}{dt} &= \frac{i_{L2}(t)d_2(t) + i_{L1}(t)\left(1 - d_2(t)\right)}{C}, \\
\frac{dv_o(t)}{dt} &= \frac{i_{L1}(t) - i_{L2}(t)}{C_o} - \frac{v_o(t)}{R_o C_o}.
\end{aligned}
\tag{5}
$$

The steady-state values of the converter variables are obtained from (5) after equating to zero the different differential equations, can be represented as follows:

$$
\begin{aligned}
V_C &= V_g, \\
D_2 &= \frac{V_o}{V_g}, \\
I_{L1} &= \frac{V_o^2}{R_o V_g}, \\
I_{L2} &= -\frac{V_o(V_g - V_o)}{R_o V_g}.
\end{aligned}
\tag{6}
$$

The converter variables can be decomposed as the same equations in (3) in the buck mode however, using $d_2$ as a duty cycle for this mode as follows:

$$d_2(t) = D_2 + \Delta d_2(t), \tag{7}$$

where is $D_2$ express the dc equilibrium point and the other term $\Delta d_2(t)$ is the small variation around it. Also in this mode, the voltage output $V_o$ should be controlled for regulating its average, and the inductor current $i_{L1}(t)$ is controlled to reshape the line current. Thus, the same two-transfer functions: the duty cycle to input inductor current $Gi_{L1}d\_d(s)$ and the duty cycle to output voltage $Gvd\_d(s)$ are expressed as:

$$Gi_{L1}d\_d(s) = \frac{E_1 s^2 + E_2 s + 2 R_o V_g A_v}{Den\_d(s) \, R_o},$$

$$Gvd\_d(s) = \frac{R_o \, V_g \, (L_s \, C \, s^2 + 1) + V_o \, s \, (L_2 - A_v \, L_s)}{Den\_d(s)},$$

where,

$$
\begin{aligned}
Den\_d(s) &= G_4 s^4 + G_3 s^3 + G_2 s^2 + G_1 s + R_o, \\
E_1 &= R_o V_g L_2 \, (C - A_v \, C_o \, (A_v - 1) + C \, C_o \, R_o \, s), \\
E_2 &= -V_o \, (L_2 \, (A_v - 1) - C_o \, R_o{}^2), \\
G_1 &= A_v{}^2 \, L_s - L_2 \, (2 \, A_v - 1), \\
G_2 &= R_o \, (C_{op} \, L_2 + C \, L_1 - A_v \, C_o \, (2 \, L_2 - A_v \, L_s)), \\
G_3 &= C \, L_1 \, L_2, \\
G_4 &= C \, L_1 \, L_2 \, C_o \, R_o.
\end{aligned}
\tag{8}
$$

### 3.3. Analysis of Numerical Results

After getting expressions for the transfer function, numerical analysis will be tested regarding to the converter parameters presented in Table 2. The input voltage maximum value is 300 V. In this numerical analysis, the voltage output varies from 200 V for buck mode and to 400 V for boost mode.

Figure 3a,b show the boost mode bode plots for the transfer functions: $Gi_{L1}d\_u(s)$ and $Gvd\_u(s)$ presented in (4) and their pole-zero mappings are shown in Figure 3c,d, respectively. As shown in the graphs, both transfer functions have RHP zero. Moreover, both transfer functions present a non-minimum phase characteristic at the high resonance undamped peaks between 2 kHz to 3 kHz. The high resonance peak can make instability by making the system oscillate and amplifying the harmonic signal at its central frequency. Additionally, while the non-minimum phase system is tricky to control, it could be controlled by decreasing the bandwidth to push the cut off frequency away from the resonance undamped frequency. Otherwise, Figure 4a,b show the buck mode bode plots for the transfer functions: $Gi_{L1}d\_d(s)$ and $Gvd\_d(s)$ presented in (8) and their pole-zero mappings are shown in Figure 4c,d, respectively. Similar to boost mode case, bode plots of the buck mode also presents a high undamped resonance frequency at almost 2 kHz. In this case, only the $Gvd\_d(s)$ presents a non-minimum phase characteristic. In both operating modes, $Gvd\_u(s)$ and $Gvd\_d(s)$ transfer functions have at least a RHP zero.

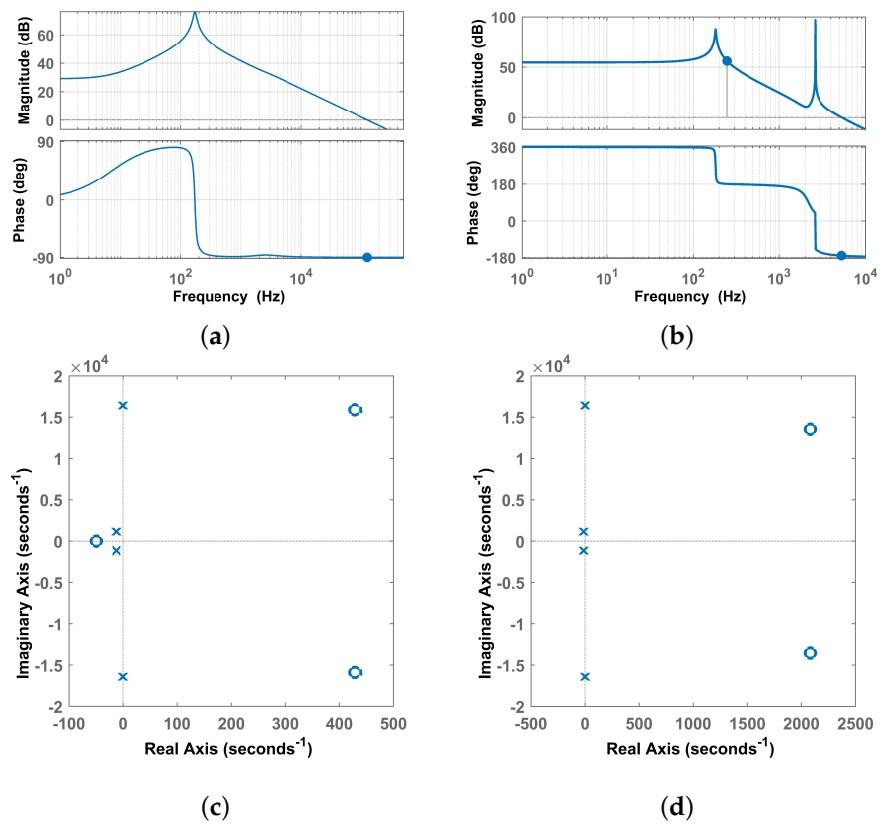

**Figure 3.** TF boost mode for (**a**) Bode plot $Gi_{L1}d\_u(s)$. (**b**) Bode plot $Gvd\_u(s)$. (**c**) Pole and zero map for $Gi_{L1}d\_u(s)$. (**d**) Pole and zero map for $Gvd$.

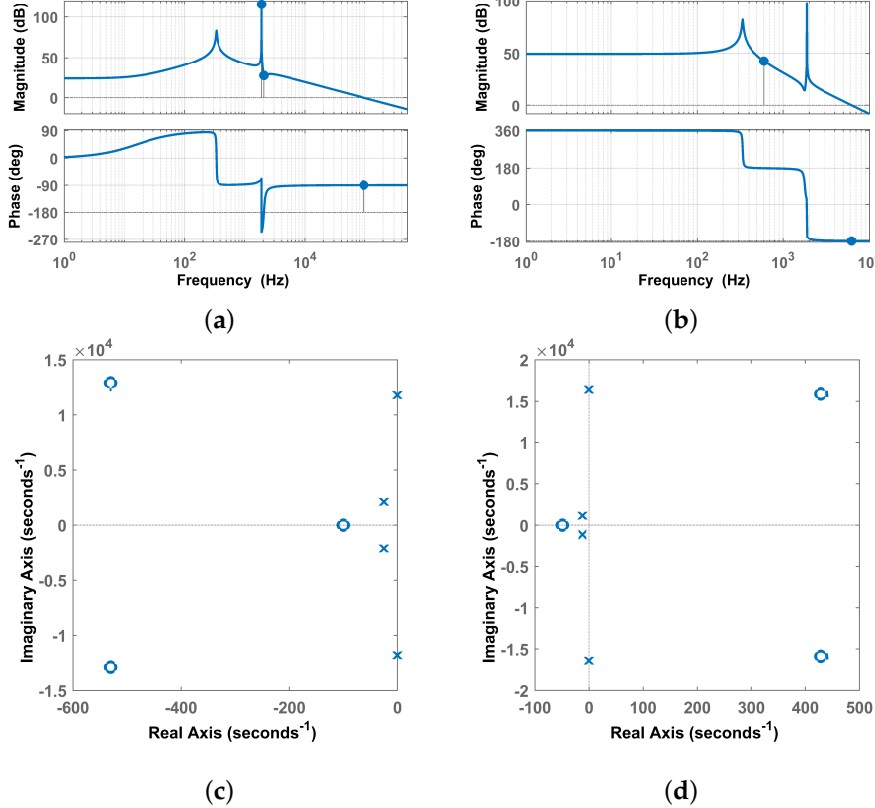

**Figure 4.** TF buck mode for (**a**) Bode plot $Gi_{L1}d\_d(s)$. (**b**) Bode plot $Gvd\_d(s)$. (**c**) Pole and zero mab for $Gi_{L1}d\_d(s)$, (**d**) Pole and zero map for $Gvd\_d(s)$.

**Table 2.** Converter parameters.

| Parameter | Value |
|---|---|
| Input voltage peak $V_g$ | 300 V |
| Input voltage frequency $F_{line}$ | 50 Hz |
| Output voltage $V_o$ | 200 V–400 V |
| Output current $I_o$ | 4–8 A |
| Switching frequency $F_s$ | 200 kHz |
| Intermediate capacitor $C$ | 8 µF |
| Output capacitor $C_o$ | 800 µF |
| Input inductance $L_1$ | 0.5 mH |
| Second inductance $L_2$ | 0.5 mH |

### 3.4. Minimum Phase System Criteria

The above numerical analysis proves that the transfer function for the current loop $Gi_{L1}d\_u(s)$ in boost mode has a RHP zero and, consequently, a non-minimum phase characteristics. Therefore, to avoid closed-loop instability, a low closed-loop bandwidth system must be designed that will decrease the PF and will increase the total harmonics distortion (THD) of the line current. The question here that will be answered in this section is "can this loop be a minimum phase by changing converter parameter?". Transfer function numerator of the current loop $Gi_{L1}d\_u(s)$ in boost mode (4) will be investigated for this purpose. It can be modeled in the 3rd order polynomial as follows:

$$P(s) \;=\; s^3 + a_2 s^2 + a_1 s + a_0. \tag{9}$$

According to the Routh–Hurwitz stability criterion, coefficients of (9) should be positive to have a stable minimum phase system. By comparing both equations in (4) and (9), the most attractive condition for stabilize this converter in boost mode can be extracted from $a_2$ coefficient and it can be expressed as follows:

$$C \;>\; C_o\left(\frac{V_o}{V_g} - 1\right), \tag{10}$$

Equation (10) is a condition for $a_2$ to be positive. Inequality in (10) is a necessary condition for boost mode transfer function $Gi_{L1}d\_u(s)$ to be minimum phase (the Routh–Hurwitz stability criterion has more conditions, and all must be accomplished to be minimum phase). Figure 5a,b shows the bode plot and its pole-zero mapping for the $Gi_{L1}d\_u(s)$ transfer function after applying the condition represented in (10). By making the output capacitor $C_o$ equal to 8 µF and intermediate capacitor C equal to 800 µF. As shown in the bode plot, the system becomes minimum phase with no RHP zero in the pole-zero map. Figure 6 shows the pole-zero map of this new converter numerator transfer function $Gi_{L1}d\_u(s)$ in boost mode with different output power. For all operating regions, the current loop is minimum phase and doesn't have a RHP zero.

### 3.5. Simulation Results

The minimum phase condition in (10) is satisfied for the boost mode, however the converter can also operate in buck mode, which needs to be validated. In circuit analysis of boost mode, the intermediate capacitor $C$ is considered as an output capacitor in this mode, so the condition of stability has no serious effect in the PF. On the other hand, in buck mode, this stability condition will deteriorate the PF, as the intermediate capacitor $C$ is an input capacitor in this mode. To prove that, the converter is simulated in PSIM using the parameters presented in Table 2. Figure 7 shows the converter current waveform $i_{L1}(t)$ and its normalized Fast Fourier Transform (FFT) plot showing the fundamental frequency

(100 Hz). Figure 7a shows a high switching frequency overlapped to the rectified grid signal. This sub-harmonic oscillation of the switching frequency appears in the FFT between 2 kHz and 3 kHz, the same range where the system has the RHP zero and the high resonance peak. This frequency appears in Figure 3a, where the high peak resonance is exactly between the same frequency range 2 kHz to 3 kHz wherein the system is non-minimum phase.

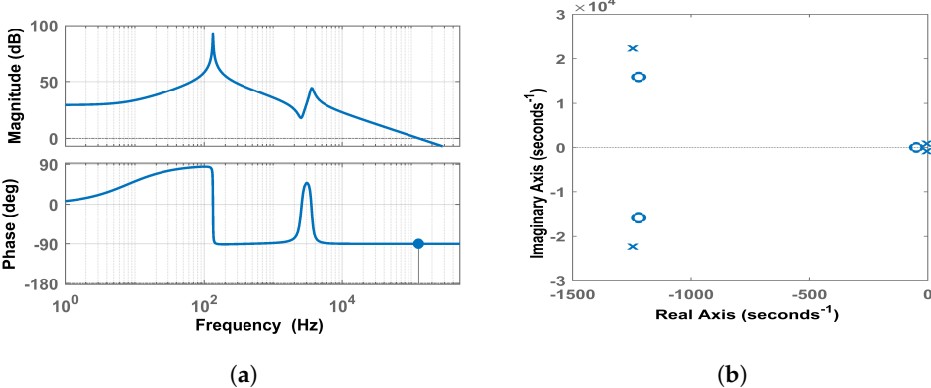

(a)                                             (b)

**Figure 5.** Transfer function $Gi_{L1}d\_u(s)$ with minimum phase condition for boost mode: (**a**) Bode plot. (**b**) Pole-zero map.

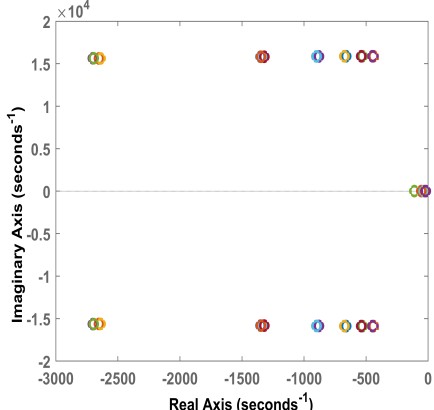

**Figure 6.** Pole and zero map of numerator $Gi_{L1}d\_u(s)$ at different output power.

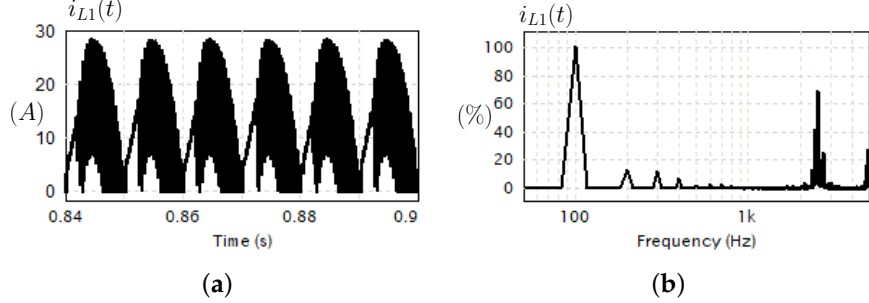

(a)                                             (b)

**Figure 7.** Inductor current $i_{L1}(t)$ in boost mode without minimum phase condition and its normalized FFT: (**a**) Inductor current waveform $i_{L1}(t)$. (**b**) Normalized FFT of Inductor current waveform $i_{L1}(t)$.

By applying the minimum phase condition in (10) for the boost mode, the system will be minimum phase and closed-loop will be stable with no RHP zero. The condition is fulfilled when the output capacitor $C_o$ is equal to 8 µF and intermediate capacitor C is equal to 800 µF. Figure 8 shows the inductor current waveform $i_{L1}(t)$ and its normalized FFT, wherein the current waveform in plot and its FFT show the absence of the previous switching frequency sub-harmonics. This validates the proposal for eliminating the RHP zero in

boost mode and the correct dynamic behavior of the converter as a PFC. Nevertheless, the feasibility of the proposal condition must be studied in the buck mode. Figure 9a shows the converter operating in buck mode with parameters not fulfilling minimum phase condition in (10). It proves that the buck mode is working, and the current shape is sinusoidal and following the reference with high PF. Figure 9b shows the inductor current waveform when the minimum phase condition in (10) is utilized. The input current waveform $i_{L1}(t)$ doesn't follow the current reference, and has a distortion factor that leads to lower PF. As observed in the simulation results, modifying the converter parameters to ensure minimum phase characteristic in boost mode of operation makes the converter not feasible as a PFC when it operates in buck mode. Therefore, another strategy must be studied to ensure minimum phase characteristic and feasible operation as PFC in both operating modes.

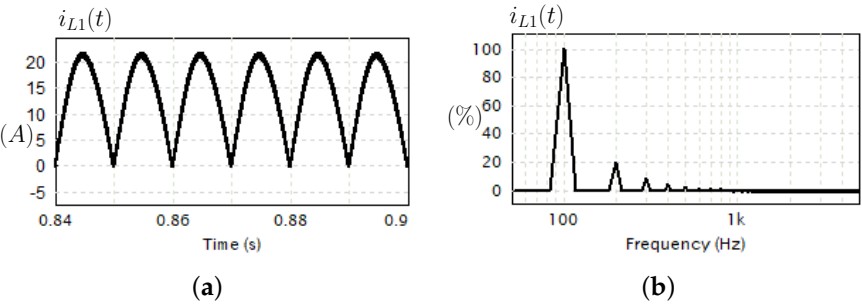

(**a**)                                                                   (**b**)

**Figure 8.** Inductor current $i_{L1}(t)$ in boost mode with minimum phase condition and its normalized FFT: (**a**) Inductor current waveform $i_{L1}(t)$. (**b**) Normalized FFT of inductor current waveform $i_{L1}(t)$.

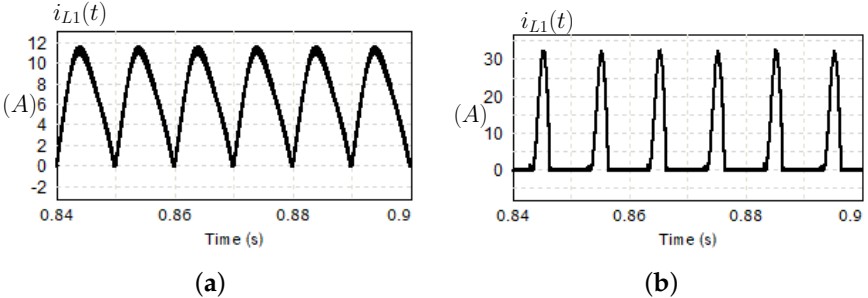

(**a**)                                                                   (**b**)

**Figure 9.** Inductor current $i_{L1}(t)$ in buck mode: (**a**) Without minimum phase condition. (**b**) With minimum phase condition.

## 4. Boost/Buck Converter Extension with Snubber RC Circuit

As discussed in the previous sections regarding to the problem of the non-minimum phase system for the current control loop, this section will discuss solving this problem by introducing RC snubber circuit to damp the resonance peak and change the system to be minimum phase in both modes. Figure 10 shows the converter after adding a damping RC ($R_d$, $C_d$) circuit parallel to the intermediate capacitor $C$. The small signal modeling will be discussed for both modes buck and boost to check the stability and operation regions.

### 4.1. State-Space Averaging and Small-Signal Model Analysis in Boost Mode with RC Snubber

State-space average equations of the converter shown in Figure 10 for the boost mode will be discussed. The state-space average equations will have another extra equation due to the RC damping components. The voltage $v_{cd}$, which is the voltage across the damping capacitor $C_d$, is changing regarding the switching cycle and should be expressed.

The state-space average equations for the boost mode with RC damping circuit can be expressed as follows:

$$
\begin{aligned}
\frac{di_{L1}(t)}{dt} &= \frac{v_g(t) - v_o(t) + d_1(t)\,v_c(t)}{L_1}, \\
\frac{di_{L2}(t)}{dt} &= \frac{-v_c(t) + v_o(t)}{L_2}, \\
\frac{dv_c(t)}{dt} &= \frac{i_{L2}(t) - d_1(t)\,i_{L1}(t)}{C} - \frac{v_c(t) - v_{cd}(t)}{R_d C}, \\
\frac{dv_{cd}(t)}{dt} &= \frac{v_c(t) - v_{cd}(t)}{C_d R_d}, \\
\frac{dv_o(t)}{dt} &= \frac{i_{L1}(t) - i_{L2}(t)}{C_o} - \frac{v_o(t)}{R_o C_o}.
\end{aligned}
\tag{11}
$$

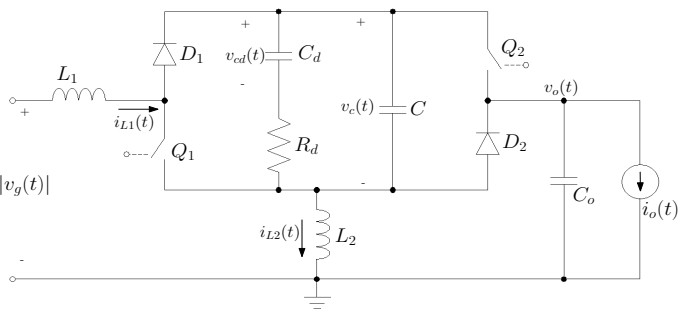

**Figure 10.** lBuck boost converter with RC snubber damping.

The DC equilibrium equations are similar to equations that were obtained in (2), with extra an equation for $v_{cd}$ that can be expressed as follows:

$$
V_{cd} = V_c = \frac{V_g}{1 - D_1}.
\tag{12}
$$

The converter variables can be decomposed the same as equations obtained in (3), adding an extra equation due to the RC snubber components as follows:

$$
v_{cd}(t) = V_{cd} + \Delta v_{cd}(t),
\tag{13}
$$

where $V_{cd}$ represents the dc equilibrium point and $\Delta v_{cd}(t)$ represents small variation around it. Two new transfer functions must be calculated with the RC snubber circuit: the duty cycle to inductor current $Gi_{L1}d\_u\_RC(s)$ and the duty cycle to output voltage $Gvd\_u\_RC(s)$ represents as follows:

$$
\begin{aligned}
Gi_{L1}d\_u\_RC(s) &= \frac{A_v^3\,V_g\left(C_o\,L_2\,R_d\,R_o\left(F_1 s^2 + F_2 s^3\right) + F_3 s + 2\right)}{Den\_u\_RC(s)}, \\
Gvd\_u\_RC(s) &= \frac{A_v^2\,V_g\left(H_1 s^3 + H_2 s^2 + H_3 s + R_o\right)}{Den\_u\_RC(s)},
\end{aligned}
$$

where,

$$
\begin{aligned}
Den\_u\_RC(s) &= A_v{}^2 L_2 (M_5 s^5 + M_4 s^4 + M_3 s^3 + M_2 s^2) + M_1 s + R_o, \\
F_1 &= \frac{C_d}{L_2}\left(\frac{C}{C_o} + 1\right) + \frac{C_d}{C_o H_4}\left(C_{dp} - (A_v - 1)\left(C_o + \frac{H_4}{R_o{}^2}\right)\right), \\
F_2 &= \left(\frac{C}{R_d} + C\,C_d\,s + \frac{C_d}{R_o}\left(\frac{C}{C_o} - A_v + \frac{R_o}{R_d} + 1\right)\right), \\
F_3 &= R_o\left(C_{op} + C_d\left(\frac{2\,R_d}{R_o} + 1\right) - \frac{L_2\,(2\,A_v - 1)}{R_o{}^2}\right), \\
H_1 &= A_v\,C\,H_4\,L_2, \\
H_2 &= \frac{A_v\,H_4}{R_o}\left(L_2\left(\frac{C_{dp}\,R_o{}^2}{H_4} + 1\right) - A_v\,L_s\right), \\
H_3 &= -L_s\,A_v{}^2 + L_2\,A_v + H_4, \\
H_4 &= C_d\,R_d\,R_o, \\
M_1 &= H_4 + A_v{}^2 L_s - L_2\,(2\,A_v - 1), \\
M_2 &= \frac{H_4}{R_o}\left(\frac{L_1}{L_2} - \frac{2}{A_v} + \frac{1}{A_v{}^2} + 1\right) + \frac{L_s\,R_o\,(C_{op} + C_d)}{L_2} - \frac{C_o\,R_o\,(2\,A_v - 1)}{A_v{}^2}, \\
M_3 &= H_4\left(C + C_o\left(\frac{1}{A_v{}^2} - \frac{2}{A_v} + 1\right) + \frac{C_{op}\,L_1}{L_2}\right) + L_1\,C_{dp}, \\
M_4 &= C_o\,L_1\,(C_{dp}\,R_o + C\,C_d\,R_d), \\
M_5 &= C\,C_o\,H_4\,L_1 \\
C_{dp} &= C + C_d.
\end{aligned}
\tag{14}
$$

### 4.2. State-Space Averaging and Small-Signal Model Analysis in Buck Mode with RC Snubber

The state-space average equations of the converter in the buck mode with damping RC shown in Figure 10 can be expressed as:

$$
\begin{aligned}
\frac{di_{L1}(t)}{dt} &= \frac{v_g(t) - v_o(t) - v_c(t)(1 - d_2(t))}{L_1}, \\
\frac{di_{L2}(t)}{dt} &= \frac{v_o(t) - v_c(t)d_2(t)}{L_2}, \\
\frac{dv_c(t)}{dt} &= \frac{i_{L2}(t)d_2(t) + i_{L1}(t)(1 - d_2(t))}{C} + \frac{-v_c(t) + v_{cd}(t)}{R_d\,C}, \\
\frac{dv_{cd}(t)}{dt} &= \frac{v_c(t) - v_{cd}(t)}{C_d R_d}, \\
\frac{dv_o(t)}{dt} &= \frac{i_{L1}(t) - i_{L2}(t)}{C_o} - \frac{v_o(t)}{R_o C_o}.
\end{aligned}
\tag{15}
$$

The DC equilibrium equations are the same obtained in (6) with an extra equation for the $V_{cd}$ that can be expressed as:

$$
V_{cd} = V_g. \tag{16}
$$

Finally, the two transfer functions: the duty cycle to inductor current $Gi_{L1}d\_d\_RC(s)$ and the duty cycle to output voltage $Gvd\_d\_RC(s)$ are represented as:

$$
\begin{aligned}
Gi_{L1}d\_d\_RC(s) &= \frac{A_v\,L_2\,V_g\left(N_2 s + \frac{2}{L_2} + \frac{1}{R_o}H_4\,N_3\,s^2 + C\,C_o\,H_4\,N_1\,V_g\,s^3\right)}{Den\_d\_RC(s)}, \\
Gvd\_d\_RC(s) &= \frac{V_g\left(P_3 + \frac{1}{R_o}P_2 s + L_1\,L_2\,P_1\,s^2 + 1\right)}{Den\_d\_RC(s)},
\end{aligned}
\tag{17}
$$

where,

$$
\begin{aligned}
Den\_d\_RC(s) &= Y_5 s^5 + Y_4 s^4 + Y_3 s^3 + Y_2 s^2 + Y_1 s + 1, \\
N_1 &= \frac{1}{V_o}\left(s + \frac{1}{C\,R_d} + \frac{1}{C_d\,R_d} + \frac{1}{C_o\,R_o} - \frac{A_v\,(A_v - 1)}{C\,R_o}\right), \\
N_2 &= \frac{2\,C_d\,R_d + C_o\,R_o}{L_2} - \frac{A_v - 1}{R_o}, \\
N_3 &= \frac{C_o\,R_o}{L_2} + \frac{1 - A_v}{R_o} + \frac{1}{A_v\,R_d} + \frac{C_o\,R_o}{H_4}\left(\frac{C}{A_v\,C_o} - A_v + 1\right), \\
P_1 &= C_{dp}\,L_P + \frac{A_v\,H_4}{R_o{}^2}\left(\frac{1}{L_1} - A_v\,L_P\right), \\
P_2 &= -L_s\,A_v{}^2 + L_2\,A_v + H_4, \\
P_3 &= \frac{C\,H_4\,L_1\,L_2\,L_P\,s^3}{R_o}, \\
L_P &= \frac{1}{L_1} + \frac{1}{L_2}, \\
Y_1 &= \frac{H_4 + L_2 + A_v{}^2\left(L_1 - L_2\left(\frac{2}{A_v} - 1\right)\right)}{R_o}, \\
Y_2 &= L_1 L_2 \left(\frac{C_{op} + C_d\left(\frac{R_d}{R_o} + 1\right)}{L_1} + \frac{C_{dp}}{L_2} + A_v{}^2\left(L_P - \frac{2}{A_v\,L_1}\right)\left(C_o + \frac{H_4}{R_o{}^2}\right)\right), \\
Y_3 &= \frac{H_4\,L_2}{R_o}\left(C_o\left(A_v{}^2 - 2\,A_v + \frac{A_v{}^2\,L_1}{L_2} + 1\right) + C\,L_1\left(L_P + \frac{C_d + C}{H_4\,C}\right)\right), \\
Y_4 &= C\,C_d\,C_o\,L_1\,L_2\left(\frac{1}{C} + \frac{1}{C_d} + \frac{R_d}{C_o\,R_o}\right), \\
Y_5 &= C\,C_d\,C_o\,L_1\,L_2\,R_d.
\end{aligned}
$$

### 4.3. Analysis of Numerical Results

After getting the expressions of $Gi_{L1}d\_RC(s)$ and $Gvd\_RC(s)$ for both modes of operations, bode plots and pole-zero maps will be analyzed. The parameters presented in Table 2 are used plus parameters of damping RC: i.e., $C_d = 100$ μF and $R_d = 7$ ohms. Figure 11a,b show the boost mode bode plots of the transfer functions $Gi_{L1}d\_u\_RC(s)$ and $Gvd\_u\_RC(s)$ represented in (14) and their pole-zero mappings are shown in Figure 11c,d, respectively. As shown in the bode plots Figure 11a, the current transfer function $Gi_{L1}d\_u\_RC(s)$ is a minimum phase system due to the damping RC circuit which damped the resonance peak. In Figure 11c, it is obvious that the complex RHP zero are pushed to be in the left half plane due to the damping effect of the RC snubber. Furthermore, the bode plot of the voltage transfer function $Gvd\_u\_RC(s)$, which is shown in Figure 11b, is minimum phase system as well.

Figure 12a,b show the buck mode bode plots of the transfer functions $Gi_{L1}d\_d\_RC(s)$, $Gvd\_d\_RC(s)$ represented in (17) and their pole-zero plots are shown in Figure 12c,d, respectively. As shown, the two transfer functions also represent minimum phase systems and have no resonance peak. From the pole-zero mappings, the plots don't have RHP zero as well. Therefore, the two modes are minimum phase systems and can be easily controlled.

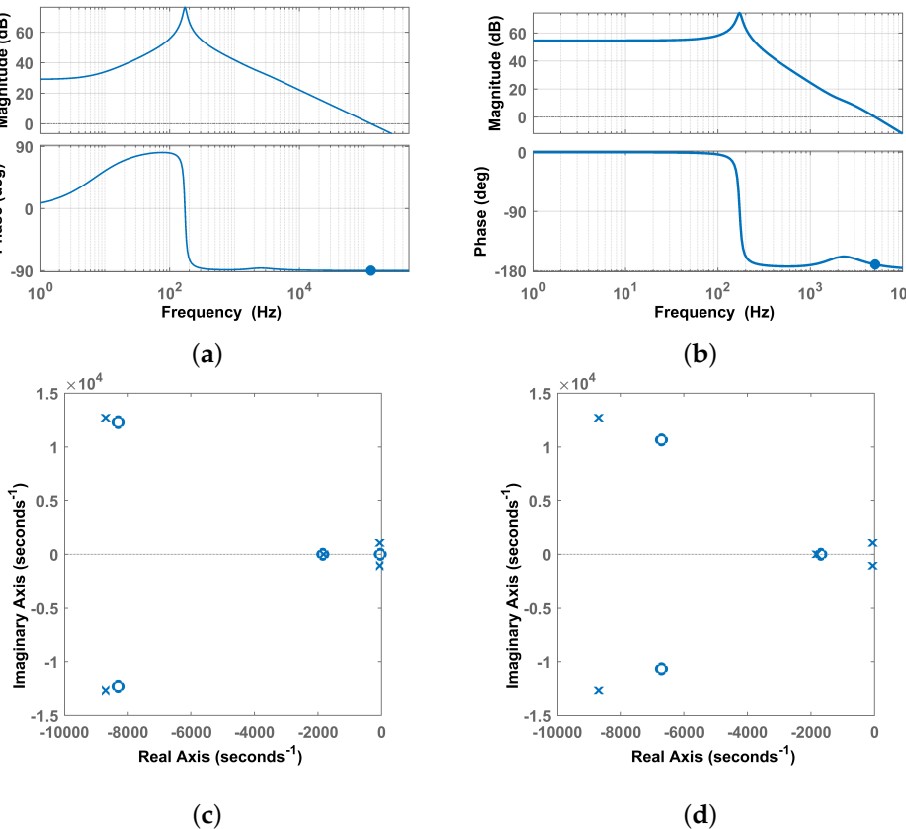

**Figure 11.** TF boost mode for: (**a**) Bode plot $Gi_{L1}d\_u\_RC(s)$. (**b**) Bode plot $Gvd\_u\_RC(s)$. (**c**) Pole and zero map for $Gi_{L1}d\_u\_RC(s)$. (**d**) Pole and zero map for $Gvd\_u\_RC(s)$.

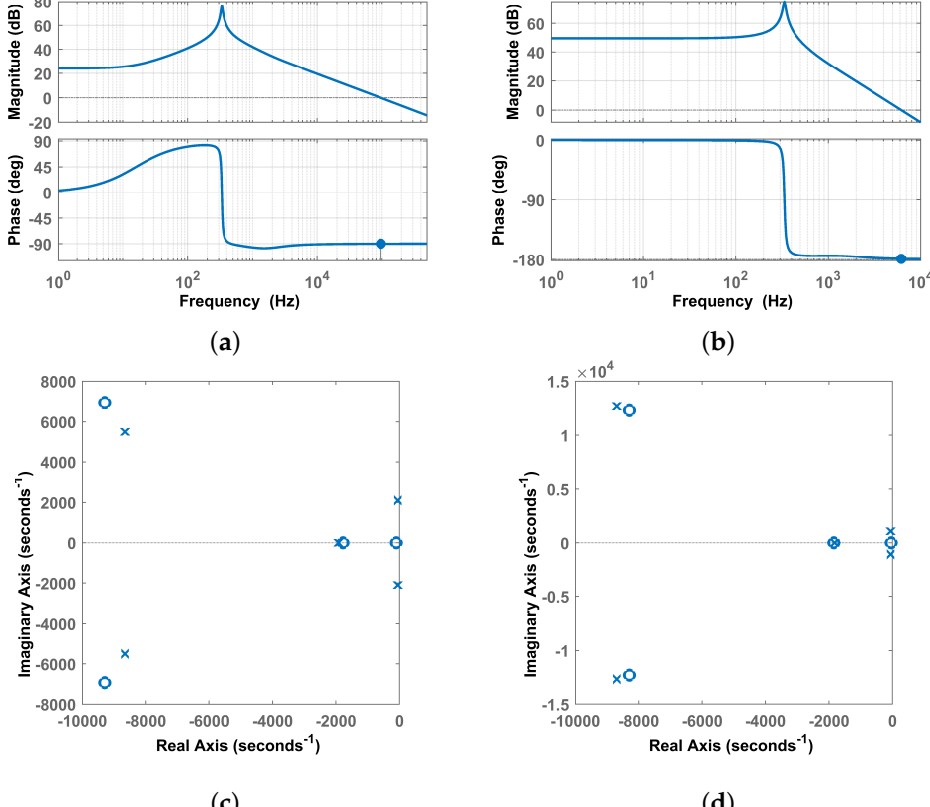

**Figure 12.** TF buck mode for: (**a**) Bode plot $Gi_{L1}d\_d\_RC(s)$. (**b**) Bode plot $Gvd\_d\_RC(s)$. (**c**) Pole and zero map for $Gi_{L1}d\_d\_RC(s)$. (**d**) Pole and zero map for $Gvd\_d\_RC(s)$.

## 5. Control Design

System block diagram of the small signal model can be shown in Figure 13. This section discusses the design of the current and voltage controller $G_{Ci}(s)$ and $G_{CV}(s)$, respectively. The converter has two loops, the inner current loop, and the outer voltage loop. Thus, the inner current loop should be designed first by design the $G_{Ci}(s)$. In order to reshape the input current and maintain a high PF for all operating modes, the loop gain of the inner current loop should have a high bandwidth and its crossover frequency should be lower than the switching frequency. Additionally, regarding robustness, it should have a phase margin higher than 45°. The loop gain for the inner, fast current loop is a good approximation at frequencies below the switching frequency ($F_s$) and around the input voltage used in calculations. This loop gain is varying with input voltage and must have good stability margins for all input voltages. In boost mode, the transfer function for the current control $G_{Ci}(s)$ can be presented as follows:

$$G_{Ci}(s) \ = \ \frac{s+2011}{s}, \tag{18}$$

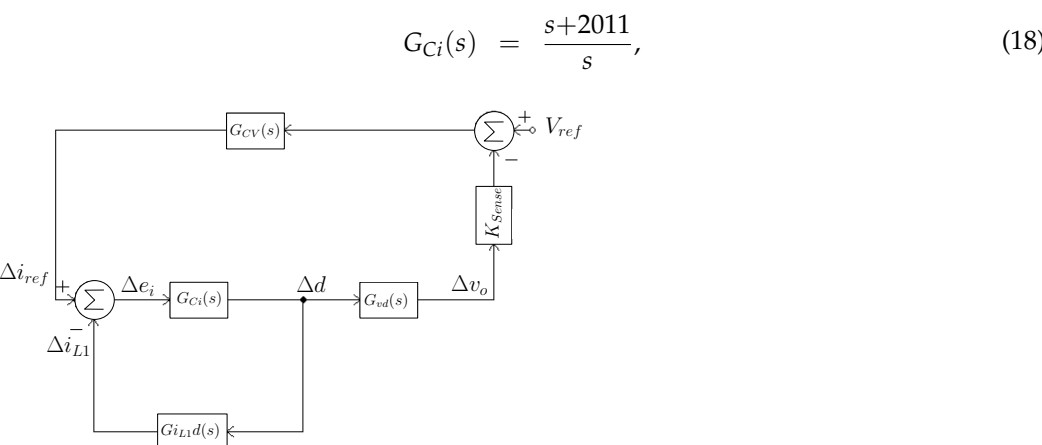

**Figure 13.** System block diagram of the small signal model.

Figure 14a represents the loop gain transfer function of the current loop. Wherein the cross over frequency is almost 100 kHz with phase margin 90°.

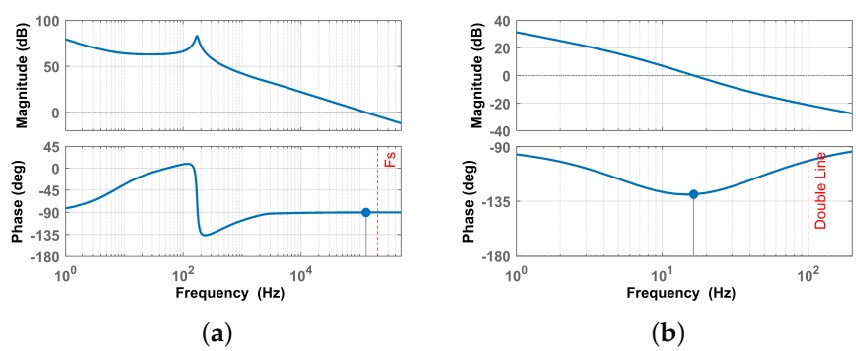

(a) (b)

**Figure 14.** Bode plots transfer functions for boost loop gain of: (**a**) Current loop $Gi_{L1}d\_u\_RC(s)$ $G_{Ci}(s)$. (**b**) Voltage loop $G_{vi_{ref}}(s)G_{CV}(s)K_{sense}$.

The closed loop transfer function $G_{i_{L1}d-CL}(s)$ for the inner current becomes:

$$G_{di_{ref}}(s) \ = \ \frac{G_{Ci}(s)}{1+G_{Ci}(s) \ \ Gi_{L1}d(s)}. \tag{19}$$

The outer, slow voltage loop is directly affected by input voltage variations at twice the line frequency (double line), it is a good approximation only at frequencies below this. Thus, the cross over frequency should be less than the double line frequency with phase margin more than 45°. Therefore, design of the output voltage control $G_{CV}(s)$ is achieved

by using (19) and (14) to obtain the current reference to output voltage transfer function $G_{vi_{ref}}(s)$ as follows:

$$G_{vi_{ref}}(s) = G_{vd}(s)G_{i_{L1}d-CL}(s). \tag{20}$$

The transfer function of the voltage control loop $G_{CV}(s)$ is designed to obtain a phase margin of 50° at a crossover frequency of 55 Hz, which can be expressed as:

$$G_{CV}(s) = \frac{0.125s + 25}{0.005s}, \tag{21}$$

The $G_{CV}(s)$ transfer function is first order with integrator and zero at low frequency. Figure 14b represents the loop gain for voltage loop transfer function, confirming the bandwidth is 55 Hz and phase margin 50°. For the control simplicity, the same $G_{CV}(s)$ and $G_{Ci}(s)$ will be used in controlling the buck mode as well. Unlike implementation in [13] for the control circuit, wherein two different controller loops are used for the boost mode and another loop for buck mode. Thus, increasing the control complexity and computation time from the digital control point of view. Figure 15a,b show the bode plots loop gain of the current and voltage transfer functions in the buck mode, respectively. Figure 15a is the bode plot of the critical loop, which is the current loop, wherein the bandwidth is 100 kHz with phase margin is almost 90°. Figure 15b is the bode plot transfer function of the voltage loop gain wherein the bandwidth is 25 Hz and the phase margin is 68°.

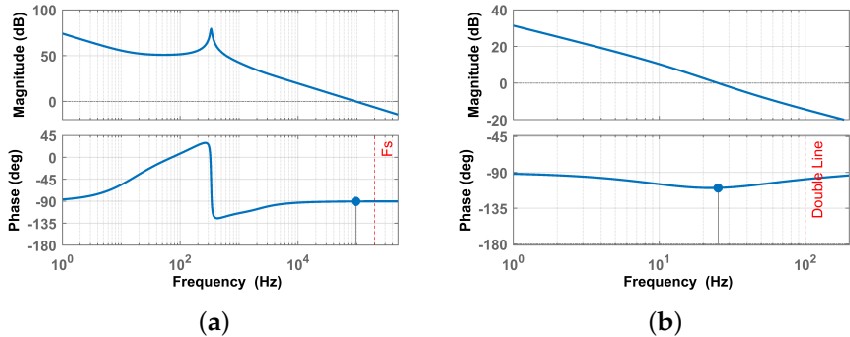

(a)  (b)

**Figure 15.** Bode plots transfer functions for buck loop gain of: (**a**) Current loop $Gi_{L1}d\_d\_RC(s)G_{Ci}(s)$. (**b**) Voltage loop $G_{vi_{ref}}(s)G_{CV}(s)K_{sense}$.

Figure 16 shows the block controller diagram. It consists of two loops, current control loop and voltage control loop. The input voltage $v_g(t)$, output voltage $v_o(t)$ and inductor current $i_{L1}(t)$ are sensed to be processed in the controller. First, a peak detector circuit is used to detect the peak of the sensed input voltage then divide the sensed input voltage by its peak to have a normalized rectified sine waveform voltage reference $v_{rect\_Ref}(t)$. The sensed output voltage is compared with a fixed reference voltage to maintain a fixed average output voltage by using the compensator $G_{CV}(s)$. Next, the output of $G_{CV}(s)$ is multiplied by the rectified sine waveform voltage reference $v_{rect\_Ref}(t)$, giving the reference current for the current controller compensator $G_{Ci}(s)$. Wherein the sensed inductor current $i_{L1}(t)$ is compared by the reference current to reshape the input current and maintain a high PF. The output of the $G_{Ci}(s)$ is compared twice with a Sawtooth waveform (1 V amplitude) using two different comparators to generate the corresponding gate control signals of $Q_1$ and $Q_2$. The top comparator generates the gate signal of switch $Q_2$ by comparing the output from the $G_{Ci}(s)$ with Sawtooth waveform, which has the designed switching frequency. The bottom comparator is used to drive the boost switch $Q_1$ by subtract 1 V from the $G_{Ci}(s)$ output voltage and then compare the result with the same Sawtooth waveform.

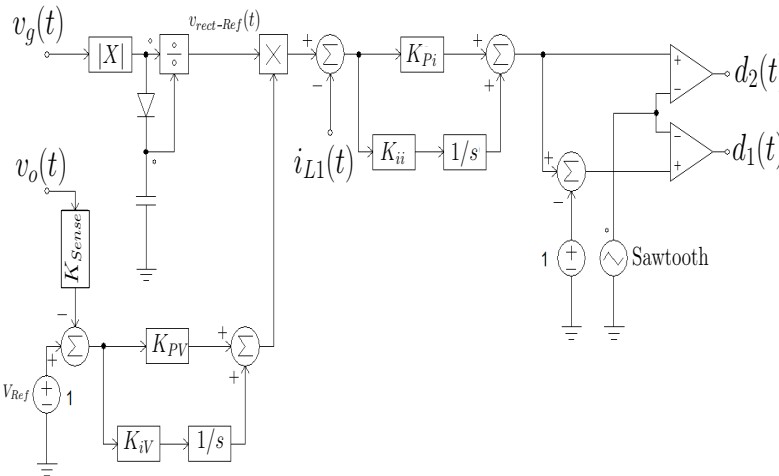

**Figure 16.** Controller circuit schematic diagram.

## 6. Simulation Results

The feasibility and performance of the proposal will be evaluated by means of simulation using PSIM software. The simulated schematic, which consists of the power stage explained in Figure 1 and the control circuit presented in Figure 16, is combined with the parameters of Table 2. To make the simulation more realistic, ESR are being added to the inductance and capacitance, the switches are Level-2 model and the diodes also include their parasitic effects. The load is a sink current source. The input is a sinusoidal input voltage, which is being rectified by a passive H-bridge. Figure 17 shows the steady state waveforms of the inductor current $i_{L1}(t)$ at boost and buck operation modes at full load after using the RC Snubber. These two waveforms can be compared with the previous Figures 8 and 9. In both operation modes, the current waveform of the inductor $L_1$ follows the reference and reshapes the current to a rectified sinusoidal waveform. Figure 18 shows the steady state waveforms of input voltage $v_g(t)$, input current $i_g(t)$, output voltage $v_o(t)$ and the output current $i_o(t)$ at two different output currents. The output voltage is 200 V and the peak input voltage is 300 V, which operates the converter in buck mode. Figure 18a shows the waveforms when the converter operates at the full load absorbing 8 A. This figure depicts how the input current waveforms perfectly follow the current reference and it is in phase with the input voltage with low distortion factor, wherein the PF is measured to be 0.99. The output voltage ripple can be controlled by the value of the output capacitance according to the application requirement, here the output capacitor is 800 µF/450 V. The output voltage ripple is 25 V, which is 12.5% of the averaged value. Figure 18b shows the waveforms when the converter operates at the half load absorbing 4 A. This figure depicts how the input current waveforms perfectly follow the current reference and it is in phase with the input voltage with low distortion factor, wherein the PF is measured to be 0.99. The output voltage ripple is 18 V which is 9% of the averaged value.

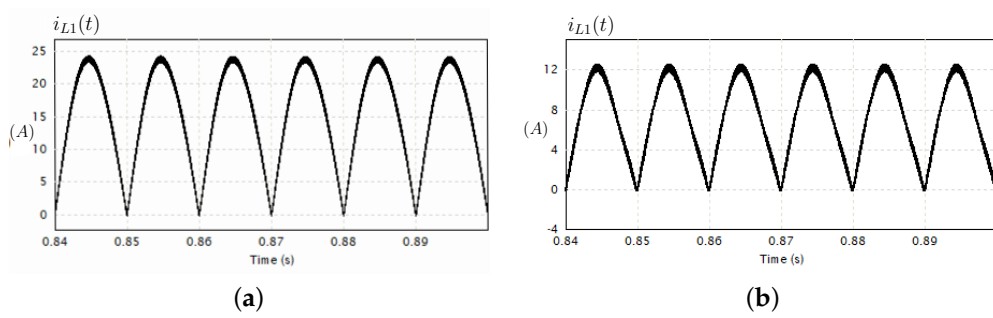

**Figure 17.** Waveforms of inductor current $i_{L1}(t)$ in: (**a**) Boost mode. (**b**) Buck mode.

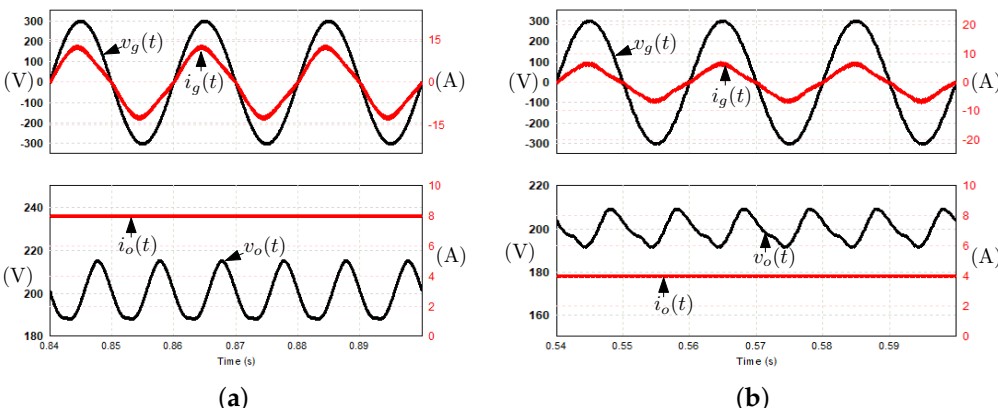

**(a)**                                                                 **(b)**

**Figure 18.** Steady state waveforms for buck mode of input voltage $v_g(t)$, input current $i_{L1}(t)$, output voltage $v_o(t)$ and the output current $i_o(t)$ with: (**a**) Full power. (**b**) Half power.

Figure 19 shows the steady state waveforms of input voltage $v_g(t)$, input current $i_g(t)$, output voltage $v_o(t)$ and the output current $i_o(t)$ at two different output currents. The output voltage is 400 V and the peak input voltage is 300 V which means the converter operates in boost mode. Figure 19a shows the steady state waveforms at full load where the current source absorbs 8 A. This figure represents how the input current waveforms perfectly follow the current reference and it is in phase with the input voltage with low distortion factor, wherein the PF is measured to be 0.99. The output voltage ripple is 30 V, which is 7.5% of the averaged value.

Figure 19b shows the waveforms when the converter operates in the half load where the current source absorbs 4 A. This figure depicts how the input current waveforms perfectly follow the current reference and it is in phase with the input voltage with low distortion factor, wherein the PF is measured to be 0.99. The output voltage ripple is 20 V which is 5% of the averaged value.

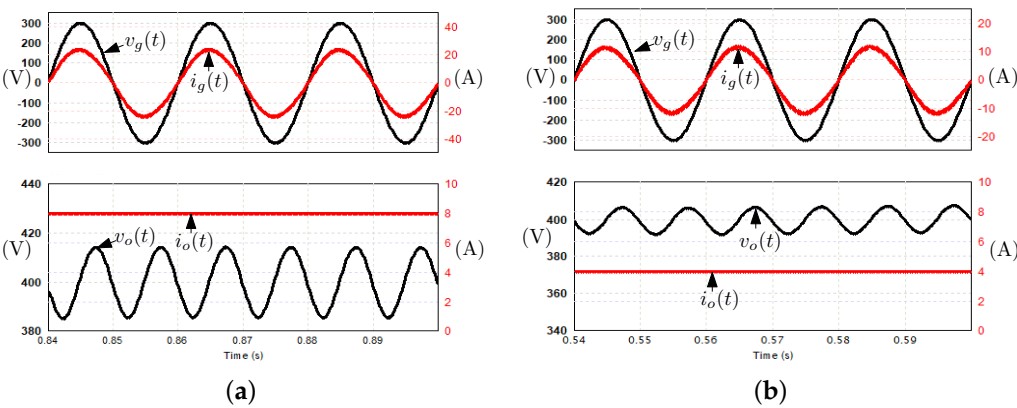

**(a)**                                                                 **(b)**

**Figure 19.** Steady state waveforms for boost mode of input voltage $v_g(t)$, input current $i_{L1}(t)$, output voltage $v_o(t)$ and the output current $i_o(t)$ with: (**a**) Full power. (**b**) Half power.

To show the dynamics of the converter controller after developing to be minimum phase system, output current transient is tested to check for this purpose. There are two mode of transient that tested for this converter by allowing the current source to step down from the high current (full power) to the low current (half power), and vice versa from the low current to the high current. Then, checking the other waveforms till they arrived in steady state point. This test is done for both converter modes boost and buck wherein, the input voltage is fixed 300 V and the output voltage changes from 200 V for buck mode to 400 V for the boost mode.

Figure 20a shows the boost mode transients waveforms for input voltage $v_g(t)$, input current $i_g(t)$, output voltage $v_o(t)$ and the output current $i_o(t)$ at two different output current values. The transient happens at 0.3 s where the current output change from full current load to half current load. Examining the output voltage and the input current, it is shown in the figure that they reach the steady state after 200 ms, which represents 10 line cycles. The other transient happens at 0.6 s where the current load change from the half current load to full current load reaching the steady state after 100 ms, which represents 5 line cycles. It should be noted here that in the first transient the time response was slower than the second transient due to the slow dynamic of the bulky output capacitor.

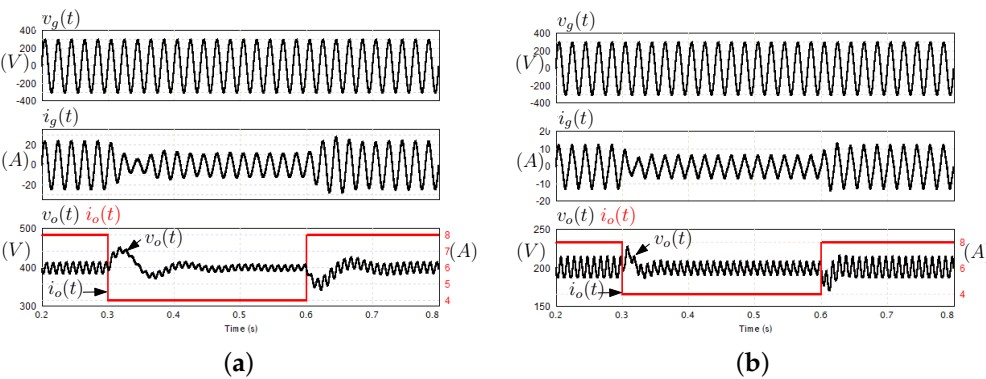

**Figure 20.** Output current transient waveforms of input voltage $v_g(t)$, input current $i_g(t)$, output voltage $v_o(t)$ and the output current $i_o(t)$ from full power to half power for: (**a**) Boost mode. (**b**) Buck mode.

The other buck mode transients waveforms for input voltage $v_g(t)$, input current $i_g(t)$, output voltage $v_o(t)$ and the output current $i_o(t)$ at two different output current values are shown in Figure 20b. The same as the boost, the first transient happens at 0.3 s, where the current output changes form full current load to half current load. Examining the output voltage and the input current, it is shown in the figure that steady state is reached after 60 ms, which represents three line cycles. The other transient happens at 0.6 s, where the current load changes from the half current load to full current load and the waveforms reached steady state after 40 ms, which represents two line cycles. it should be noted that the snubber circuit will decrease the total converter efficiency, due to the power loss in the resistance Rd. Therefore, a correct choice of Rd and Cd values must be done in order to achieve minimum-phase characteristic minimizing the effects on the efficiency.

## 7. Conclusions

This paper presents the analysis and control of a four switches boost/buck AC/DC converter for PFC applications with a wide output voltage range. The duty cycle to input current transfer function in mode boost and the duty cycle to output voltage in both modes of operation present a RHZ, thus implying a limitation in the closed-loop operation, slower dynamics and poor performance parameters. Different modifications in converter design and its parameters are analyzed and tested to overcome this limitation. Finally, an RC damping network eliminates the RHZ limitation, allowing a wider closed-loop bandwidth operation. A unique control is designed for both modes of operation, showing fast transient behavior, high power factor and low harmonic distortion. Theoretical predictions are validated by simulated results using PSIM.

**Author Contributions:** Conceptualization, M.N., E.V.-I. and J.C.; methodology, M.N.; software, M.N.; validation, M.N.; formal analysis, M.N.; investigation, M.N.; resources, M.N.; writing—original draft preparation, M.N.; writing—review and editing, M.N., E.V.-I. and J.C.; visualization, M.N.; supervision, E.V.-I. and J.C.; project administration, J.C. All authors have read and agreed to the published version of the manuscript.

**Funding:** The research was supported by Universitat Rovira i Virgili and Diputacion de Tarragona under grant Marti Franques 2019 PMF-PIPF-95.

**Institutional Review Board Statement:** Not applicable.

**Informed Consent Statement:** Not applicable.

**Data Availability Statement:** Data is contained within the article.

**Conflicts of Interest:** The authors declare no conflict of interest.

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
