# Peer review of "Analysis of Non-Minimum Phase System for AC/DC Battery Charger Power Factor Correction Converter"

_applsci, doi:10.3390/app12020868_

Round 1
Reviewer 1 Report
This paper presents a deep analysis of the model for a four-switch buck-boost PFC. The minimum phase condition was summarised and case studies for both operation modes were carried out. Then, an RC snubber is added to extend the minimum phase condition based on small-signal modelling. Simulation studies were used to support the statements.
Fig 17 b) shows the current waveform has higher distortion than boost mode. Please comment on the role of controller design for optimising such performance.
Author Response
Dear Reviewer,
We want to express our gratitude to you for your comments, suggestions, and your valuable time.
Authors

Reviewer 2 Report
1.In this paper a four-switch buck-boost based AC/DC converter is analyzed by using the well known state space averaging technique. Similar to the basic boost converter, due to the transfer time delay of the stored energy in the inductor, the duty ratio to output voltage transfer function of the concerned converter also contains right half plane zeros which will reduce the closed loop bandwidth of the system and degrade the converter performance , namely power factor and total harmonic distortion. The major contribution of this paper lies in proposing adding a damping RC series circuit in parallel with the intermediate capacitor to eliminate the RHP zeros of the transfer functions. Then a two loop controller is designed to demonstrate the effectiveness.
2. There are some mistakes that should be corrected in the context. For example, equations (4) and (8) are not correct. The transfer functions should be a 4th order system. However, Den-u(s) and Den-d(s) are only with order 3. As can also be observed from Fig.3 and Fig.4, there are four poles. The models should be revised.
3. Similarly, by adding the parallel RC damping branch, the minimum phase system now becomes a 5th order system. However, equation (17) is with 4th order. One can also see from Fig.11, there are five poles. The model should be revised.
4. Only simulation results are provided in the paper. For power electronic systems, normally a prototype should also be constructed to verify the validity. In addition, The starting transient as well as light loading condition should be covered. Finally, normally resistance should not be used in the converter topology to achieve high efficiency , the authors should comment the influence of adding the damping resistor on the converter efficiency so that the value of the proposed strategy can be judged.
Author Response
Dear Reviewer,
We want to express our gratitude to you for your comments, suggestions, and your valuable time.
Regards,
Authors

Reviewer 3 Report
Dear Authors,
The submitted article relates to such an important area of research as improving chargers. The results obtained are of both scientific and practical interest. I have a few recommendations that I hope will help improve the manuscript before publication:
1. The introduction should be supplemented with a specific research contribution to the subject area.
2. Several comments can be added why linearization of equations (1) is required.
3. What numerical method was used to solve ODE systems in numerical simulations?
4. The font size in fig. 3-6, 11, 12, 14, 15 should be increased.
5. How were the constants chosen in formulas (18), (21)?
6. Has the proposed control law been investigated from the point of view of guaranteeing the stability of the numerical solution?
7. The list of references can be expanded by research of recent decades. In particular, some of the early works of the authors can be mentioned.
Author Response

(The authors gave the same response as above.)
